# Experimental violation of *n*-locality in a star quantum network

Davide Poderini[1], Iris Agresti[1], Guglielmo Marchese [1], Emanuele Polino[1], Taira Giordani[1], Alessia Suprano[1], Mauro Valeri[1], Giorgio Milani[1], Nicolò Spagnolo[1], Gonzalo Carvacho[1], Rafael Chaves[2] & Fabio Sciarrino [1✉]

The launch of a satellite capable of distributing entanglement through long distances and the first loophole-free violation of Bell inequalities are milestones indicating a clear path for the establishment of quantum networks. However, nonlocality in networks with independent entanglement sources has only been experimentally verified in simple tripartite networks, via the violation of bilocality inequalities. Here, by using a scalable photonic platform, we implement star-shaped quantum networks consisting of up to five distant nodes and four independent entanglement sources. We exploit this platform to violate the chained *n*-locality inequality and thus witness, in a device-independent way, the emergence of nonlocal correlations among the nodes of the implemented networks. These results open new perspectives for quantum information processing applications in the relevant regime where the observed correlations are compatible with standard local hidden variable models but are nonclassical if the independence of the sources is taken into account.

[1] Dipartimento di Fisica, Sapienza Università di Roma, P.le Aldo Moro 5, I-00185 Roma, Italy. [2] International Institute of Physics & School of Science and Technology, Federal University of Rio Grande do Norte, 59078-970, P. O. Box 1613, Natal, Brazil. ✉email: fabio.sciarrino@uniroma1.it

**B**ell's theorem[1], i.e., the incompatibility of quantum predictions with local hidden variable (LHV) models, is among the most influential results in quantum foundations. Despite being almost 60 years old, only recently the phenomenon of Bell nonlocality has been proven in a loophole-free manner in a series of independent experiments[2–6]. Apart from its fundamental importance, witnessing Bell nonlocality is at core of many applications in quantum information processing, ranging from quantum communication[7] and distributed computing[8], to quantum cryptography[9], quantum key distribution[10,11], and randomness generation[12,13].

Indeed, the violation of a Bell inequality allows to bound the secure key rate that can be exchanged among distant nodes in a device-independent (DI) manner[14–16], that is, with minimal assumptions on measurement apparatuses and relying exclusively on observed data. Most of these findings, however, only hold for the paradigmatic bipartite Bell scenario. Generalizations to the multipartite case have been proposed, for instance, based on multipartite entangled states, such as the Greenberger-Horne-Zeilinger states (GHZ)[17]. Notwithstanding, generation of such states on a photonic platform is still highly demanding[18], affecting the near-term experimental relevance of device-independent communication protocols based on it. Accordingly, quantum networks of growing size and complexity aiming at the so-called quantum internet[19,20] are much more likely to be composed of independent sources, each one generating small size entangled states but at much higher quality and rate.

The independence of the sources in a quantum network gives rise to a much richer set of correlations[21–30] as compared with the standard Bell nonlocality. Not only the network scenario allows for nonlocality activation[31] and less stringent detection efficiencies[32], but also the emergence of truly new kinds of non-classical behaviors[30]. In spite of much recent theoretical advance, experimental implementations[6,33–35] have been so far limited to the simplest possible quantum network, the bilocality model[21,22]. Such scenario is akin to the entanglement swapping experiment, involving two independent sources of entangled states, and only very recently was implemented while closing the locality and independence loopholes at the same time[36].

In this work, we provide a proof of principle demonstration of the scalability of such networks, moving beyond the bilocality scenario, by considering up to four independent sources distributing entanglement among five nodes in a star-network topology. To ensure the independence of the sources, we employ four photonic setups located in four different laboratories, each one consisting of a source of entangled photons, pumped by a different laser, and a measurement station. Via this platform, we report the violation of polynomial chained Bell inequalities with an increasing number of parties and measurement settings, hence detecting the presence of nonlocal correlations among the nodes of multipartite networks. Our scalable approach could be useful both for demonstrating the nonlocality of topologically different scenarios[30], as well as a testbed for device-independent protocols of information processing in quantum networks[37].

## Results

**Causal modeling approach.** Topologically complex networks, composed of several sources and parties, are notoriously hard to characterize, especially if the independence of the sources is taken into consideration. A new approach has been introduced in the last few years to properly analyze these networks, which leverages on the theory of causal inference[38]. Indeed, the causal modeling approach[39] has become a customary and powerful tool to represent classical and quantum networks[27,40–43]. In this framework, directed acyclic graphs (DAGs) provide both a graphical and a mathematical way to visualize causal structures (see Fig. 1). Each of the nodes stands for a random variable involved in the process, whose cause and effect relations are encoded by directed edges (arrows).

**Bilocality scenario.** The simplest quantum network beyond the paradigmatic Bell scenario (Fig. 1a) is the so-called bilocality scenario[21,22] depicted in Fig. 1b, that is, a network with two independent sources $\Lambda_1$ and $\Lambda_2$. The three nodes, Alice, Bob, and Charlie, can perform measurements of different observables denoted by $x_1$, $x_2$, and $y$, chosen independently, with outcomes labeled as $a_1$, $a_2$, and $b$, respectively. Note that such independence in the choice of the measurements is also crucial in standard Bell tests[6], in particular for the derivation of Bell inequalities[22,27,41].

The aforementioned experiment is described by a probability distribution that, under the bilocality assumption, should be

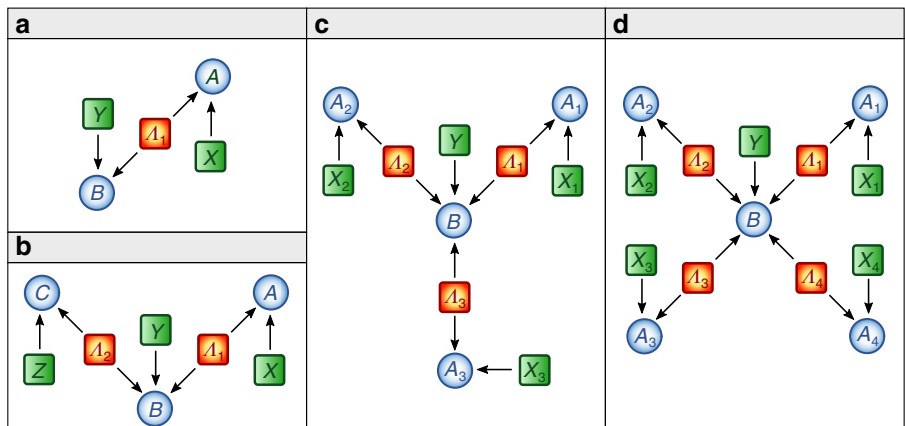

**Fig. 1 Directed acyclic graph (DAG) representation for the star scenario with different number of sources. a** Causal structure of the standard Bell scenario in which a single source $\Lambda_1$ mediates the correlations between the two measurement stations with outcomes $A$ and $B$ and measurement choices $X$ and $Y$, respectively. **b** DAG for the bilocal scenario where two independent hidden variables $\Lambda_1$ and $\Lambda_2$ distribute the correlations to three measurement stations with outcomes $A$, $B$, and $C$ and choice of measurements $X$, $Y$, and $Z$, respectively. **c, d** DAG for the star-network with three and four independent sources $\Lambda_1$, $\Lambda_2$, $\Lambda_3$, and $\Lambda_1$, ... , $\Lambda_4$, respectively. Each non-central measurement station has outcomes $A_i$ (measurement choices $X_i$) and the central has outcome $B$ (measurement choices $Y$).

compatible with a bilocal LHV model, given by refs. [21,22]

$$p(a_1, a_2, b|x_1, x_2, y)$$
$$= \sum_{\lambda_1, \lambda_2} p(\lambda_1)p(\lambda_2)p(a_1|x_1, \lambda_1)p(a_2|x_2, \lambda_2)p(b|y, \lambda_1, \lambda_2). \quad (1)$$

Assuming that all these variables are dichotomous (with values 0 or 1), any correlation compatible with the bilocal model (1) should fulfill the following nonlinear Bell inequality:

$$S = \sqrt{|I_1|} + \sqrt{|I_2|} \leq 1, \quad (2)$$

with

$$I_1 = \frac{1}{4} \sum_{x_1, x_2 = 0,1} \langle A_1^{x_1} A_2^{x_2} B^0 \rangle, \quad (3)$$

$$I_2 = \frac{1}{4} \sum_{x_1, x_2 = 0,1} (-1)^{x_1 + x_2} \langle A_1^{x_1} A_2^{x_2} B^1 \rangle, \quad (4)$$

and $\langle A_1^{x_1} A_2^{x_2} B^y \rangle$ being the expectation value of the measurements outcomes of the three nodes:

$$\langle A_1^{x_1} A_2^{x_2} B^y \rangle = \sum_{a_1, a_2, b = 0,1} (-1)^{a_1 + a_2 + b} p(a_1, a_2, b|x_1, x_2, y). \quad (5)$$

In the last 2 years, inequality (2) has been experimentally violated in a number of photonic setups[6,33–36]. The first violation[33] exploited polarization-entangled photons and the measurements performed in the central node $B$ relied on a complete Bell-state measurement. Notably, bilocality violation can also be achieved with separable measurements[22,23], as experimentally realized in refs. [6,35]. The violation of bilocality is relevant, in particular, in the intermediate situation where local variable models can reproduce the observed correlations but bilocal models cannot[33], namely one can violate a bilocal causality inequality even if the data admits a LHV model, where the independence of the sources is not taken into account. Accordingly, quantum states generating classical correlations in conventional scenarios can become powerful resources in a network, thus enlarging the capabilities to process information in a non-classical way.

**$n$-locality in a star-network**. The bilocality scenario can be extended to more complex networks, considering an increasing number of independent sources and involved nodes, as well as different topologies. In particular, a $n$-locality scenario involves $n$ independent sources distributing correlations among the nodes. A topology that has received much theoretical attention is the star-network[21–25], with $n + 1$ nodes interconnected by $n$ independent sources, which contains the bilocality scenario as a particular case (see Figs. 1b–d). Each of the $n$ peripheral nodes (called $A_i$) is connected through the source $\Lambda_i$ to the central node of the network (called $B$). Labeling the measurements for $B$ and external nodes $A_i$ by $y$ and $x_i$ and their outcomes by $b$ and $a_i$, respectively, a classical $n$-local hidden variable model implies that any classical observed distribution should be decomposable as

$$p(a_1 \ldots a_n b|x_1 \ldots x_n y)$$
$$= \sum_{\lambda_1, \ldots \lambda_n} p(a_1|x_1 \lambda_1) \ldots p(a_n|x_n \lambda_n)p(b|y\lambda_1 \ldots \lambda_n)p(\lambda_1) \ldots p(\lambda_n).$$
$$(6)$$

Considering that each of the nodes can perform $k$ measurements described by dichotomous observables $B^y$ (central node) and $A_i^{x_i}$ (external nodes), the $n$-local model implies the following

nonlinear Bell inequality[23]:

$$S_n^k = \sum_{i=1}^{k} |I_i|^{1/n} \leq k - 1 \quad (7)$$

where $I_i = \frac{1}{2^n} \sum_{x_1, \ldots, x_n = i-1}^{i} \langle A_1^{x_1} \cdots A_n^{x_n} B^{i-1} \rangle \quad (8)$

with $A_i^k = -A_i^0$. This is a generalization both of bilocality inequality (2), as well as of the so-called chained Bell inequality[44], which are recovered in the case of $n = 2$ and $n = 1$, respectively. For this reason, we call inequality (7) the chained $n$-locality inequality. A larger number of measurement settings $k$ for the chained form considered here provides advantages in device-independent protocols, either by reducing the experimental constraints for their violation[45] or by leading to better security tests[46]. The set of classical correlations allowed by $n$-local models is shown in Fig. 3.

Let us now focus on the quantum violation of these inequalities, by highlighting a few important points. First, inequality (7) makes the explicit assumption that the sources of correlations are independent, thus any implementation of the corresponding causal structure should take that into account. For instance, the first experimental implementations[33,34] adopted a single laser in order to generate the two pairs of entangled states, opening a loophole in the quantum violation of inequality (2). To avoid this loophole, as detailed below, in our experimental implementation, we used independent entanglement sources in separated laboratories. The second important point is that, in the star-network, Bob (the central node) has access to $n$ independent physical systems, thus the most general measurement he can perform is a measurement in an entangled basis. In fact, theoretical results showing the activation of nonlocality in such networks[31] rely on measurements on a GHZ basis. From the experimental perspective, however, such entangled measurements represent an extremely demanding task, as even measurements on a complete Bell basis cannot be implemented using linear optics, without resorting to hybrid or nonlinear approaches[47,48]. Nicely, however, it has been shown that the optimal quantum violation of the $n$-locality inequality can already be achieved if all measurements, including those of the central node in the networks, are made in separable bases[22–24], thus avoiding the request to synchronize photons from different pairs as needed for projection on Bell basis. In summary, not only one can rely on independent sources but also make the simplest possible measurements to detect nonlocality in complex networks, paving the way of an experimentally scalable approach.

Let us now quantify the quantum violation of the $n$-locality inequality. By generating singlet entangled states and by performing the measurements reported in the Methods section, the upper bound of inequality (7) is the following:

$$S_n^k = k \cos(\pi/2k), \quad (9)$$

that can be shown to be the optimal quantum violation[24]. We note that such bound does not depend on the number of nodes, $n$, but only on the number of measurements settings, $k$.

To experimentally implement the star-network causal structure depicted in Fig. 1d, we exploited the photonic platform of Fig. 2 with four independent polarization-entangled photon pair sources (a detailed description of the sources can be found in Supplementary Note 1). As shown in Fig. 2, three of the sources, that are pumped by different pulsed lasers located in the laboratories 1, 3, and 4, rely on spontaneous parametric down-conversion (SPDC) of type II, achieved through Beta-Barium Borate crystals, which emits degenerate photon pairs at 785 nm.

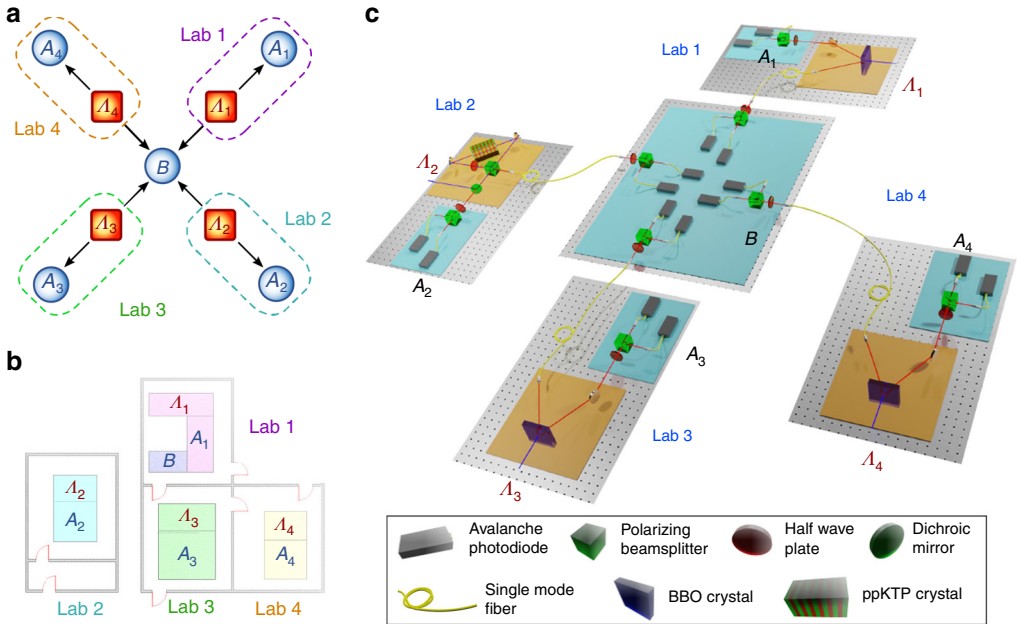

**Fig. 2 Experimental apparatus. a** Four independent polarization-entangled photon pair sources and five measurement stations are available for the experimental realization of violation of the chained $n$-locality inequality (7) in a star-network configuration. **b** Physical location of the laboratories and scheme of the experimental apparatus. Distinct laboratories contain one source and one measurement station each, with the exception of Lab. 1, which also contains the central node $B$ of the star-network on a separate optical table. **c** The entangled photon sources $\Lambda_1$, $\Lambda_3$, and $\Lambda_4$ are realized using a Beta-Barium Borate (BBO) crystal, pumped in a pulsed regime, which emits photon pairs at 785 nm using spontaneous parametric down-conversion (SPDC) of type II. The source $\Lambda_2$ instead is pumped in continuous-wave regime and employs a periodically poled KTP crystal placed inside a Sagnac interferometer to generate entangled photon pairs at 808 nm via a type II SPDC process. Single photons are then measured in polarization using a half-wave plate (HWP) followed by a polarizing beam splitter (PBS).

Instead, the fourth source, located in laboratory 2, is pumped in the continuous-wave regime and generates photon pairs at 808 nm. This source scheme exploits a SPDC process of type II in a periodically poled KTP nonlinear crystal, placed into a Sagnac interferometer[49]. Thus, in our experimental implementation of the star-network (Fig. 1d), we strongly enforce the independence of the sources for the implemented causal structure.

When a pair of photons is generated in each crystal, it is split: one photon is sent to the central node $B$, while the other remains within the laboratory $A_i$ where it was generated, to be measured locally. The distribution of the shared photons among the nodes was accomplished by exploiting different optical fiber links. The maximum length of the fibers used was 25 m. At the output of each fiber, the polarization of the photon was properly compensated, to counteract unavoidable rotations along the path. In each laboratory, there is a measurement station, where polarization analysis can be performed by rotating a half-wave plate (HWP) placed right before a polarizing beam splitter (PBS). This apparatus allows each party to perform a number $k$ of different projective measurements on their subsystem, since they are all represented by linear combinations of the Pauli matrices $\sigma_x$ and $\sigma_z$. Accordingly, to perform the $k$ measurement settings required to test inequality (7), each HWP was rotated to switch up to four different settings, in order to perform all the possible measurement combinations, which, in our star-shaped network, amount to 1024 combinations.

Photons arrival events were registered by single photons detectors (SPD) and a different time-tagger in each laboratory, and immediately sent to a central station where the synchronization and the coincidence detection took place. In particular, two detected photons were considered as simultaneous, provided that they were registered within a time, shorter than a given window. The results presented here were obtained using a coincidence

time window of 80 μs, but a quantum violation was also observed for narrower values, up to 0.49 μs (see Supplementary Note 3).

We carried out the experiments for causal structures of star networks scenarios with $n = 2, 3, 4$ sources, whose independence is enforced by exploiting different independent pump lasers for SPDC processes occurring in dislocated laboratories, as mentioned above. Let us note that our setup relies, inevitably, on the measurement independence assumption, i.e., free choice of inputs measurements, as well as on the lack of correlations among the sources. Both such assumptions, indeed, cannot be enforced by any physical principle, but only be made less plausible as possible. For this reason, as already mentioned, we enforced the independence of the sources by adopting different laser devices, which, furthermore, were not relying on the same electric power source. Moreover, it requires the fair-sampling assumption, analogously to other most relevant experiments dealing with quantum networks[6,33–36]. In the end, given that the measurements were not performed providing space-like separation, we are assuming the lack of uncontrolled communication channels among the measurements stages, located in the separated laboratories.

In this framework, we managed to distinguish synchronous events by designing a sophisticated software, keeping track of the generation times of all of the four sources, as detailed in the Supplementary Note 3. On the other hand, by not requiring photons of different sources to interfere, we offered the possibility of using different types of sources within the same network and to have significantly higher generation rates.

### Violations of $n$-locality with $k = 2$ measurement settings.
The scheme previously described allows for the violation of $n$-locality with $k$ different measurement settings. When considering the

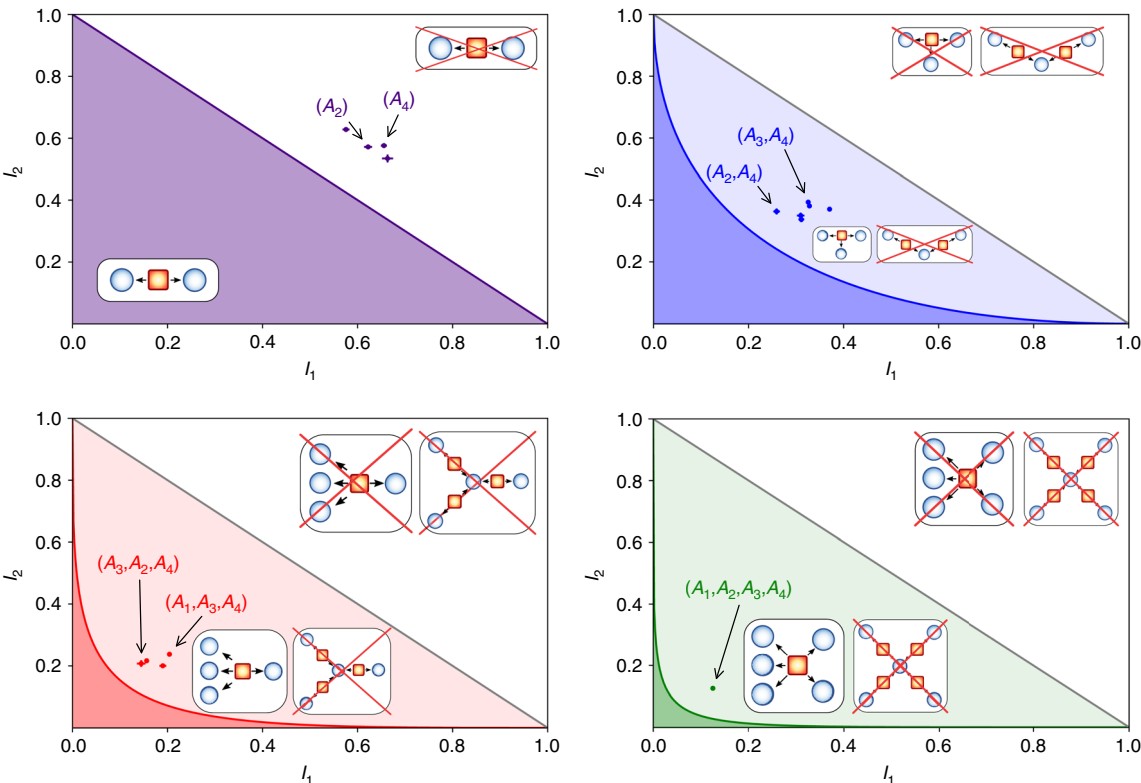

**Fig. 3 Experimental correlations represented in the $I_1$, $I_2$ plane ($k = 2$ settings).** The purple, blue, red, and green regions represent the classical correlations for 1-, 2-, 3-, and 4-local scenarios, respectively. The 2-local case corresponds to the bilocal inequality while the 1-local reduces to the paradigmatic CHSH inequality. For $n \geq 2$, the gray line bounds the correlations allowed by a local model with no assumption on sources independence. The green point represents the $I_1$, $I_2$ values measured in the case of $n = 4$ sources. The red, blue, and purple points represent the experimental values for all the possible combination of laboratories which are 4, 6, and 4 in the 3-, 2-, and 1-star cases, respectively. Error bars represent the standard error of the mean taken over different sequential acquisitions.

particular case of $k = 2$, the optimal measurement settings to perform at the external nodes and summarized in Eq. (10), are performed by the HWPs at measurement stages before PBS and with angles $\theta_0^A = 0°$ for choice $x_i = 0$ and $\theta_1^A = 22.5°$ for choice $x_i = 1$. For the central node $B$, the HWP's angles that project along states in Eq. (13) are $\theta_0^B = 11.25°$ for choice $y = 0$ and $\theta_1^B = 33.75°$ for choice $y = 1$. The corresponding results for this case are presented in Fig. 3 and can be extended by increasing the number of measurements $k$. The data in Fig. 3 show that the observed values are incompatible with the classical models for $n$-locality. Nonetheless, the obtained correlations are still compatible with a LHV model with a single shared source among all parties. To witness the non-classicality of the network correlations, the source independence has to be taken into account. In particular, for $n = 2$, we measured such parameters exploiting all of the six combinations of pairs among the four sources. The maximum observed value is $S_{max}^{obs} = 1.218 \pm 0.002$, violating the classical bound by 109 standard deviations. For $n = 3$, our setup allows for four triples of sources, achieving $S_{max}^{obs} = 1.199 \pm 0.004$, a violation of 50 standard deviations with respect to the classical bound. Finally, for $n = 4$ we obtained a value $S_{max}^{obs} = 1.192 \pm 0.005$, corresponding to a violation of 38 standard deviations. All the results for $k = 2$ are reported in Table 1.

**Violations of $n$-locality with $k > 2$ measurement settings.** An increasing number of settings provides advantages in DI protocols based on the violation of Bell inequalities[37]. For that reason, we also consider the violation $n$-locality chained inequality (7)

with an increasing number $k$ of settings. In particular, we violated the 4-locality ($n = 4$) with up to four different measurement settings corresponding to a total 1024 combinations. Using this apparatus, we were able to violate the $n$-locality chained inequality (7) for the 2-, 3-, and 4-star-shaped networks scenarios, counting the 4-, 6-, and 8-fold coincidences events, in the case of $k = 2, 3, 4$ measurement settings. For the case of $n = 2$ and $n = 3$, each of these measurements was performed on every pair and triplet of the different four parties, thus allowing for six and four different combinations, respectively. The results for the different cases are presented in Fig. 4 as well as Table 2. For the particular case of $n = 4$ with $k = 4$, our results yields $S = 3.157 \pm 0.002$, surpassing the classical limit by 71 standard deviations. The experimental values are fully compatible with the theoretical predictions where the noise of the whole system is taken into account (see Supplementary Note 2). Albeit with our apparatus we cannot close the locality loophole, because we would require an increase of the spatial distance among the parties, we approached such condition as much as possible through a reduction of the time window for the detection of coincidence events. Indeed, we present results also for narrower coincidence windows, up to 0.49 μs, which still allow for significant quantum violations (see Supplementary Note 3). Such reduction of the coincidence window lowers the probability of mutual causal influences among the parties in realistic experimental conditions.

## Discussion
Bell's theorem is a conceptual cornerstone in our understanding of quantum theory that, with the rise of quantum information

**Table 1 Experimental results for different number of sources $n$ and $k = 2$ measurement settings.**

| $n$ Sources | Combination | $I_1$ | $I_2$ | $S^{obs}$ |
|---|---|---|---|---|
| 1 | $A_1$ | 0.576 ± 0.007 | 0.628 ± 0.002 | 1.204 ± 0.009 |
| | $A_2$ | 0.622 ± 0.008 | 0.572 ± 0.006 | 1.197 ± 0.010 |
| | $A_3$ | 0.662 ± 0.011 | 0.534 ± 0.012 | 1.194 ± 0.010 |
| | $A_4$ | 0.655 ± 0.006 | 0.576 ± 0.004 | 1.232 ± 0.007 |
| 2 | $A_2, A_4$ | 0.259 ± 0.007 | 0.363 ± 0.010 | 1.111 ± 0.011 |
| | $A_3, A_4$ | 0.325 ± 0.004 | 0.393 ± 0.004 | 1.198 ± 0.005 |
| | $A_3, A_2$ | 0.310 ± 0.008 | 0.350 ± 0.010 | 1.147 ± 0.011 |
| | $A_1, A_4$ | 0.3279 ± 0.0017 | 0.381 ± 0.003 | 1.190 ± 0.003 |
| | $A_1, A_2$ | 0.311 ± 0.006 | 0.337 ± 0.009 | 1.138 ± 0.010 |
| | $A_1, A_3$ | 0.370 ± 0.003 | 0.371 ± 0.003 | 1.217 ± 0.003 |
| 3 | $A_3, A_2, A_4$ | 0.145 ± 0.009 | 0.207 ± 0.010 | 1.116 ± 0.015 |
| | $A_1, A_2, A_4$ | 0.156 ± 0.005 | 0.217 ± 0.007 | 1.139 ± 0.009 |
| | $A_1, A_3, A_4$ | 0.204 ± 0.005 | 0.238 ± 0.005 | 1.208 ± 0.006 |
| | $A_1, A_3, A_2$ | 0.190 ± 0.007 | 0.200 ± 0.008 | 1.160 ± 0.010 |
| 4 | $A_1, A_2, A_3, A_4$ | 0.125 ± 0.005 | 0.125 ± 0.005 | 1.190 ± 0.008 |

The table shows the experimental values of $I_1$, $I_2$, and $S^{obs}$ for each possible combination of parties $\{A_1, ..., A_4\}$.

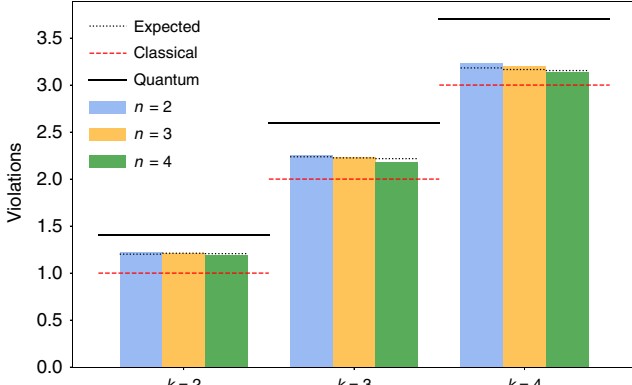

**Fig. 4 Experimental violation of the chained Bell inequality (7) for a 2, 3, and 4 sources $n$, depicted in different colors, and 2, 3, and 4 measurement settings $k$.** Solid and dashed lines represent the classical and quantum bounds in (7), respectively, while dotted lines represent the expected value of the violation for noisy states. Measurement errors are not visible in the plot, numerical values are summarized in Table 2.

science, has also been turned into a novel tool for information processing. In spite of the key developments over the years in tasks such as quantum cryptography, self-testing, and entanglement certification, generalizations of Bell's theorem beyond the simple single source scenario, in particular from the experimental point of view, are still an almost uncharted territory. That is precisely the aim of this work, to provide the first experimental implementation of a scalable quantum network consisting of an increasing number of independent entanglement sources, separated laboratories and measurement settings.

Here, we chose to investigate a topology that has received considerable theoretical attention lately, the star-network[21–25], with a central node sharing an entangled state with $n$ other peripheral nodes. In particular, we focused on the crucial task of certifying, in a device-independent manner, the presence of non-classical correlations among the nodes of the network. To this aim, we violated the chained $n$-locality inequality, an extension of the famous chained inequalities[44] to the network case. The main strengths of this inequality are its adaptability to consider an increasing numbers of nodes in the network and of measurement settings for each party and also the possibility of achieving maximum quantum violation with separable measurements, a significant experimental advantage. We violated the chained $n$-

locality inequality, thus ensuring the presence of truly non-classical correlations, by considering networks up to five nodes (four independent sources) and that each of the nodes could measure up to four different measurement settings.

In particular, networks of this kind might prove relevant, in the future, for the execution of multipartite protocols such as secret sharing[50–52]. Such task consists in a central node aiming to securely share a message with all the other parties, even if some of the receivers are untrusted. The crucial point, in this work, lies in the experimental realization of a quantum network of five stations, within which, the simultaneous presence of nonlocal correlations is certified. In this context, even though the ideal condition of space-like separated parties is not reached, we approximate such condition, by taking coincidence counts within a small time interval. Let us note that the limits on such interval are given by the rates of the sources. This approximate simultaneity, together with the independent sources, can certainly be considered as a very versatile tool for the future implementation of quantum networks and makes our platform easily adaptable to scenarios like the ones proposed by Lee and Hoban[37], that could be applied to cryptographic tasks. Furthermore, for practical purposes and applications, the short time interval, that we adopt to detect simultaneous events, makes eventual communications among the parties less plausible in a plethora of realistic scenarios. To conclude, the novelty of our work stems from the presentation of the prototype of a scalable quantum network, going beyond the bilocality scenario, which allowed, for the first time, the violation of the chained $n$-locality inequalities. Moreover, the versatility of our platform can be exploited to study networks that are attracting growing attention, like the triangle network[30] and the linear chain topology underlying quantum repeaters[53].

## Methods

**Measurements performed.** All of the four sources generate polarization-entangled pairs of photons in the Bell state $|\psi^-\rangle = (|01\rangle - |10\rangle)/\sqrt{2}$. To obtain the maximum quantum violation of inequality (7), $k\cos(\pi/2k)$, all the peripheral parties $A_i$ must perform the following projective measurements on their subsystem:

$$\left|\Psi_{k,x}^0\right\rangle = \cos(x\pi/2k)|0\rangle + \sin(x\pi/2k)|1\rangle, \tag{10}$$

$$\left|\Psi_{k,x}^1\right\rangle = \cos(x\pi/2k)|1\rangle - \sin(x\pi/2k)|0\rangle, \tag{11}$$

for each setting $x_i = x$. In turn, the central node $B$ measures each of its $n$ subsystems in the local basis:

**Table 2 Experimental results for different number of sources $n$ and measurement settings $k$.**

| $n$ Sources | $k$ Settings | $S^{obs}$ | Classical | Violation $\sigma$ | $S^{sim}$ | $S^Q$ |
|---|---|---|---|---|---|---|
| 2 | 2 | $1.217 \pm 0.003$ | 1 | 72 | $1.201 \pm 0.007$ | 1.41 |
|   | 3 | $2.253 \pm 0.002$ | 2 | 127 | $2.237 \pm 0.010$ | 2.60 |
|   | 4 | $3.2261 \pm 0.0014$ | 3 | 162 | $3.182 \pm 0.014$ | 3.70 |
| 3 | 2 | $1.208 \pm 0.006$ | 1 | 35 | $1.211 \pm 0.005$ | 1.41 |
|   | 3 | $2.227 \pm 0.003$ | 2 | 76 | $2.225 \pm 0.009$ | 2.60 |
|   | 4 | $3.195 \pm 0.002$ | 3 | 97 | $3.165 \pm 0.013$ | 3.70 |
| 4 | 2 | $1.190 \pm 0.008$ | 1 | 24 | $1.207 \pm 0.005$ | 1.41 |
|   | 3 | $2.177 \pm 0.005$ | 2 | 35 | $2.218 \pm 0.008$ | 2.60 |
|   | 4 | $3.135 \pm 0.004$ | 3 | 34 | $3.154 \pm 0.012$ | 3.70 |

The values of $S^{obs}$, $S^{sim}$, and $S^Q$ are the observed, the expected, and the maximum quantum violation, respectively. $S^{sim}$ has been computed using the state visibility estimated by Bell violations performed in each single source.

$$\left| \Phi_{k,y}^0 \right\rangle = \cos\frac{(2y+1)\pi}{4k}|0\rangle + \sin\frac{(2y+1)\pi}{4k}|1\rangle, \qquad (12)$$

$$\left| \Phi_{k,y}^1 \right\rangle = \cos\frac{(2y+1)\pi}{4k}|1\rangle - \sin\frac{(2y+1)\pi}{4k}|0\rangle, \qquad (13)$$

for each setting $y_i = y$, where the index $i$ refers the system Bob shares with the $i$-th non-central part. The resulting measurement corresponds to $B_y = B_y^1 \otimes \cdots \otimes B_y^n$, where $B_y^j$ represent the measurement performed on each subsystem. Hence, to evaluate the quantum violation in (7), we need to perform $k2^n$ combinations of measurement settings, $2^n$ for each term $I_i$ appearing in (7).

**Experimental details**. For the experimental setups of Fig. 2, the three different pump lasers for sources 1, 3, and 4, with $\lambda = 397.5$ nm are produced by a second harmonic generation (SHG) process from a Ti:Sapphire mode-locked laser with repetition rate of 76 MHz. Photon pairs entangled in the polarization degree of freedom are generated exploiting type-II SPDC in 2-mm-thick beta-barium borate (BBO) crystals. Source 2, instead, employs a continuous-wave diode laser with wavelength of $\lambda = 404$ nm, which pumps a 20-mm-thick periodically poled KTP crystal inside a Sagnac interferometer, to generate photon pairs using a type-II degenerate SPDC process. The photons generated in all the sources are filtered in wavelength and spatial mode by using narrow band interference filters and single-mode fibers, respectively.

**Coincidence counting**. The photon detection events were collected and timed by a different time-tagger device for each party, located in the corresponding laboratory (see Fig. 2b). For each 1 s of data acquisition the events were sent to a central server, along with a random clock signal shared between all the time-taggers, which was used to synchronize the timestamps of events relative to different devices. To filter out part of the noise the raw data was first pre-processed by keeping only double coincidence events for each photon source, using a narrow coincidence window of 3.24 ns. Then coincidence events between multiple sources were counted every time one of such double coincidence event was recorded for each source in a window of 80 µs.

## Data availability
The data that support the findings of this study are available from the corresponding author upon request.

## Code availability
All the custom code developed for this study is available from the corresponding author upon request.

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

## Acknowledgements

We thank F. Andreoli for useful discussions. This work was supported by The John Templeton Foundation via the grant Q-CAUSAL No. 61084, by MIUR (Ministero dell'Istruzione, dell'Università e della Ricerca), via project PRIN 2017 "Taming complexity via QUantum Strategies a Hybrid Integrated Photonic approach" (QUSHIP) Id. 2017SRNBRK, by Sapienza Progetto di Ateneo H2020-ERC QUMAC (MAChine learning via hybrid integrated QUANTUM photonics), by the Regione Lazio program "Progetti di Gruppi di ricerca" legge Regionale n. 13/2008 (SINFONIA project, prot. n. 85-2017-15200) via LazioInnova spa, and by the QuantERA ERA-NET Cofund in Quantum Technologies 2017 project HiPhoP (High dimensional quantum Photonic Platform, projectID 731473). R.C. also acknowledges the Brazilian ministries MEC and MCTIC, CNPq (Grants No. 307172/2017-1 and No. 406574/2018-9 and INCT-IQ) and the Serrapilheira Institute (grant number Serra-1708-15763). G.C. thanks Becas Chile and Conicyt.

## Author contributions

D.P., I.A., G.C., R.C., and F.S. Conceived the experiment; D.P., I.A., G.M., E.P., T.G., A.S., M.V., G.M.I., N.S., G.C., and F.S. devised and performed the experiment; D.P., G.M., I.A., E.P., N.S., G.C., R.C., and F.S. performed the data analysis; all the authors discussed the results and contributed to the writing of the paper.

## Competing interests
The authors declare no competing interests.
