## [Peer Review File · Nature Communications]

Reviewers' Comments:

Reviewer #1:

Remarks to the Author:

I thank the authors for their detailed responses.

I believe that all comments have been satisfactorily addressed.

I recommend direct publication.

Reviewer #2:

Remarks to the Author:

The authors gave satisfactory responses to my previous review (for Nature Photonics) and, in my opinion, those of the other reviewers. I believe this work is suitable for publication in Nature Communications.

Reviewer #3:

Remarks to the Author:

After reading the response letter and the revised manuscript, I would like to summarize my opinions on this new version:

1. First, I am glad to see that the coincidence window reduced to 0.49 microseconds can still lead to significant violations. This seems to me an excellent experimental achievement.

2. Compared with previous relevant works (including the work by the same group), current work indicates the following new achievements: Use several independent lasers to address multipartite cases with a relevant narrow time window (0.49 microseconds); Present a more feasible way for implementation in larger networks; Expand the network to five nodes with four sources.

3. The key method of using single-qubit measurement instead of entangled measurements in this work, which is claimed by the authors as "pivotal for the success and scalability of our experiment", has been previously experimentally realized by the same group. I think this largely reduce the novelty of the current work.

In view of these points, considering the contribution such a work could possibly bring to the "n-local" inequality field, I would like to prudently recommend its publication in Nature Communications.

Point-by-point response to the Reviewers

Reviewer #1:

*I thank the authors for their detailed responses.
I believe that all comments have been satisfactorily addressed.
I recommend direct publication.*

We thank the Reviewer for their work and for recommending direct publication of our manuscript.

Reviewer #2:

The authors gave satisfactory responses to my previous review (for Nature Photonics) and, in my opinion, those of the other reviewers. I believe this work is suitable for publication in Nature Communications.

We thank the Reviewer for their work and for recommending publication of our manuscript.

Reviewer #3:

After reading the response letter and the revised manuscript, I would like to summarize my opinions on this new version:

- 1. First, I am glad to see that the coincidence window reduced to 0.49 microseconds can still lead to significant violations. This seems to me an excellent experimental achievement.*
- 2. Compared with previous relevant works (including the work by the same group), current work indicates the following new achievements: Use several independent lasers to address multipartite cases with a relevant narrow time window (0.49 microseconds); Present a more feasible way for implementation in larger networks; Expand the network to five nodes with four sources.*
- 3. The key method of using single-qubit measurement instead of entangled measurements in this work, which is claimed by the authors as “pivotal for the success and scalability of our experiment”, has been previously experimentally realized by the same group. I think this largely reduce the novelty of the current work.*

In view of these points, considering the contribution such a work could possibly bring to the “n-local” inequality field, I would like to prudently recommend its publication in Nature Communications.

We thank the Reviewer for their positive feedback and for recommending publication in Nature Communications.